# A clinical audit on longer-term stroke management as a specific service in a primary care setting: Assessing adherence of service and clinical parameters

**Gadaffi Mostapha, Noor Azah Abd Aziz** **\*, Mohd Fairuz Ali**

Department of Family Medicine, Long-Term Stroke Clinic, HCTM UKM Primary Care Clinic, Universiti Kebangsaan Malaysia, Jalan Yaacob Latiff, Cheras, Kuala Lumpur, Malaysia

\* azah@ppukm.ukm.edu.my

## Abstract

Primary-care long-term stroke care service offers comprehensive management at the community level. A clinical audit was carried out to assess the services of this clinic as compared to the established standardized criteria for longer-term stroke care.A retrospective audit was performed to evaluate the adherence to service parameters based on eleven criteria adapted from the Canadian Post-Stroke Checklist. The following clinical parameters were audited using the Malaysian CPG on Cardiovascular Disease 2017 and Malaysian CPG on Ischemic Stroke 2020: systolic blood pressure (SBP), diastolic blood pressure (DBP), low-density lipoprotein cholesterol (LDL-C), HbA1c, and weight, and smoking status. 113 registered patients from the 2022 clinic census were audited using paired Student's *t*-test and McNemar's test.Overall, only 2 out of 11 criteria for service parameters did not meet the standard: inquiring about patient fatigue (43.4%) and access to community resources (26.5%). The attainment of the target for BP and HbA1c meets the standards set for this audit. Patients experienced reduced SBP, DBP, LDL-C, and HbA1c levels, and a statistically significant reduction was observed in DBP (4.15 mmHg, p<0.05) and LDL-C (0.30 mmol/L, p<0.05). A notable reduction in the percentage of smokers (p<0.05) was also seen.Post-stroke patients at a specific-service clinic within the primary care setting benefited from clinicians' high adherence to clinical guidelines, observed from improved clinical parameters. These may serve as an impetus for clinicians to include long-term stroke service as a specialized service within primary care specialties.

## Background

Stroke remains a major public health issue in Malaysia and many other developing countries. In Malaysia alone, the National Health Mobility Survey (NHMS) has reported increased stroke prevalence from 0.3% to 3.4% from 2006 to 2019 [1,2]. The introduction of thrombolysis management for hyperacute stroke and dedicated stroke units in acute stroke management have

**Data Availability Statement:** The authors confirm that the data supporting the findings of this study

are available within the article and/or its supplementary materials.

**Funding:** The authors received no specific funding for this work.

**Competing interests:** The authors have declared that no competing interests exist.

improved the outcome of some stroke survivors. Nonetheless, the number of stroke survivors with disability-adjusted life years (DALY) is expected to increase due to higher life expectancy, especially among the elderly, and the increased incidence of cardio-metabolic risk factors, especially diabetes and hypertension among the population in developing countries [3].

Living with stroke now includes the management of co-morbid cardiovascular risk factors, post-stroke complications, assessment of cognitive and emotional issues, and secondary prevention of stroke. Thus, it is critical not only for healthcare providers to identify and manage these issues but also to have standardized tools for assessments and planning the appropriate management plans for individual stroke survivors. Research and management on stroke focused on early phases of stroke recovery. Nonetheless, with the advancement of early stroke intervention and the success of stroke units, the focus should also be moved to the stroke survivors who live beyond hospital discharge and back to the community.

Long-term Stroke Clinic (LTSC) was established in September 2008 to provide specialized services for long-term stroke patients at Universiti Kebangsaan Malaysia (UKM) Primary Care Clinic. It was initiated as a continuation of the comprehensive treatment for stroke patients from the acute (hospital), sub-acute (hospital and interventional rehabilitation) to long-term (community, interventional rehabilitation, and preventive treatment) stages. It is operated by specialists, medical officers, and trained nurses. LTSC receives referrals from various departments at Hospital Canselor Tuanku Muhriz (HCTM) UKM (Department of Neurology, Department of Medicine, Department of Medical Rehabilitation), UKM Primary Care Clinic, and nearby clinics. As Malaysia's first comprehensive community-based stroke care service, the LTSC is part of the kick-start solution in providing comprehensive care for stroke patients after discharge. With the establishment of a post-stroke checklist as standardized parameters for post-stroke care, it is timely that the services of this clinic be assessed in terms of adherence to this checklist. Thus, a clinical audit was conducted to assess adherence to services compared to the standardized criteria for longer-term stroke care. We also audited the achievement of clinical outcomes in terms of the clinical improvement related to long-term stroke care based on standardised local guidelines.

## Methods

This retrospective audit was conducted from 1 January to 31 March 2023 at a university hospital's primary care clinic (UKM Primary Clinic, HCTM UKM) in Cheras District, Kuala Lumpur, Malaysia. This study used a universal sampling study in which all patients registered in the 2022 LTSC census and attending the clinic from 1.1.2022 until 31.12.2022 who fulfilled the inclusion and exclusion criteria were audited in the study. There were 113 patients, 68 male and 45 female patients, 50.4% Malay, 44.2% Chinese, 3.5% Indian, and 1.8% other ethnicity. The selection of the 2022 LTSC census was based on the fact that it occurred after the period of COVID-19 pandemic, during which the clinic had already resumed operations in a manner similar to that before the onset of the pandemic. The inclusion criteria were adults (aged 18 years and above at the time of diagnosis), recorded diagnosis of stroke (clinically or radiologically) which was confirmed by a medical practitioner, or a combination of these methods, or patients who attended the clinic follow-ups at least twice after registration with the clinic. Patients diagnosed with transient ischemic attack, isolated nerve palsy, or irretrievable medical records and missing data were excluded from the audit.

### Assessment tools

All relevant data were accessed through the medical records, laboratory, and pharmacy data systems. They were recorded into an audit form that contains four sections: (i) demographic

details, (ii) clinical profiles, (iii) LTSC Post-stroke Checklist (service parameters), and (iv) clinical outcome parameters. Demographic details of the patients include age, gender, ethnicity, education level, occupational status, and living arrangements. Clinical profiles of patients consisted of the age of stroke, smoking status, stroke episode, stroke subtype, presence of diabetes mellitus/ hypertension/ dyslipidaemia, and referral to neurorehabilitation program/physiotherapy/occupational/speech and language therapy post-stroke. This audit assessed the measures of the care process through adherence to service parameters based on the LTSC Post-stroke Checklist. The LTSC Post-stroke service was adapted from the Canadian Stroke Best Practices Post-Stroke Checklist [4] with two additional items (financial support and community resources) to fit the local settings. Table 1 shows the differences between Canadian Stroke Best Practice and the local checklist (Long-Term Stroke Amended Checklist) and the specific criteria listed in each checklist.

The second part of the audit involved evaluating the clinical parameters of the patients receiving the service using parameters obtained from two standardized local guidelines (Malaysian Clinical Practice Guidelines on Primary & Secondary Prevention of Cardiovascular Disease 2017 and Malaysian Clinical Practice Guidelines on Management of Ischemic Stroke 2020) [5,6]. The parameters measured are blood pressure (systolic and diastolic), glycosylated hemoglobin (HbA1c), low-density lipoprotein cholesterol (LDL-C), weight, and smoking status. To assess the performance of each clinical parameter, we defined and established the standard levels of performance for individual parameters for comparison with the audited clinical parameters.

**Table 1. Post-stroke checklist (comparison between LTSC post-stroke checklist & canadian stroke best practice post-stroke checklist).**

| ITEMS | LTSC POST-STROKE CHECKLIST | CANADIAN STROKE BEST PRACTICES POST-STROKE CHECKLIST (4) |
|---|---|---|
| 1. | Secondary Prevention<br>Medical advice/Medications | Secondary prevention |
| 2. | Physical<br>a. ADL–mRS/MBI/IADL<br>b. Mobility<br>c. Spasticity/stiffness/contracture<br>d. Pain<br>e. Incontinence | Activities of Daily Living (ADL)<br>Mobility<br>Spasticity<br>Pain<br>Incontinence |
| 3. | Speech<br>• Communication | Communication |
| 4. | Cognition assessment<br>a. Memory–ECAQ/MMSE<br>b. Emotional functioning–TQWHQ/PHQ | Cognition<br>Mood |
| 5. | Life After Stroke<br>a. Leisure activities<br>b. Driving<br>c. Back to work | Life After Stroke |
| 6. | Relationship<br>• Personal/Family | Personal Relationships |
| 7. | Fatigue | Fatigue |
| 8. | Other challenges | Other Challenges |
| 9. | Nutritional (*additional*) | |
| 10. | Financial (*additional*)<br>• Self/Social Welfare | |
| 11. | Community Resources (*additional*) | |

## Defining standards for service and outcome parameters

The standard levels of performance for each of the audited criteria were set after discussions with the topic-based experts with resources from the latest published standards of care [7,8] and a study by Al-Salti et al. [9]. Adherence to service parameters on the LTSC Post-stroke Checklist was determined when the healthcare provider assessed or mentioned the specific parameter in at least one of two consecutive visits during the audited period. Regarding adherence to service parameters, achieving 50% was determined when at least half of the patients demonstrated adherence to the service parameters outlined in Table 1. Since this was the first audit conducted on long-term stroke services, the standard levels of performance for service parameters was set at achieving a minimum of 50% for each assessed parameter. This benchmark will serve as a guide for future reassessment of these selected parameters. As for clinical outcome parameters, the percentage achieved in each outcome parameter indicates the proportion of patients who successfully attained the targeted parameters based on standard performance. The individualized standard performance level was based on the Quality Assurance (QA) indicator within the Malaysian National Diabetes Registry Report [7] and Al-Salti et al. [9] as follows: 70% for BP, 30% for HbA1c, and 45% for LDL-C. The standard performance levels for each clinical outcome parameter can be found in Table 5.

## Statistical analysis

All data were analyzed using IBM$^{®}$ SPSS$^{®}$ Statistics version 28. Kolmogorov–Smirnov and Shapiro-Wilk tests revealed that the data were normally distributed. The demographic details, clinical profiles, and service parameters were described using descriptive analysis in frequency, mean, and standard deviation (SD) as appropriate. The Student's Paired t-test and McNemar tests were used for the inferential analysis of the outcome parameters.

## Ethical statement and consent to participate

The ethical approval for this study was granted from the Research Ethics Committee of Universiti Kebangsaan Malaysia (FF-2022-157). This study was registered with the National Medical Research Register (NMRR)(RSCH ID-23-02079-WMQ, NMRR ID-23-01463-DAZ). The confidentiality of every patient was upheld throughout this research undertaking, wherein each patient was assigned an anonymized ID when their data was transferred from medical records to audit forms, and this confidentiality was maintained until the study's publication.

## Results

A total of 161 patients registered in the 2022 LTSC census were included for analysis after fulfilling the inclusion criteria. After reviewing the exclusion criteria, 29 were excluded due to irretrievable medical records or missing data, thus leaving a total of 113 patients for audit. The patients were between 41 and 88 years old, with a mean age of 64.82 years ± 9.77. Male patients (60.2%) outnumbered female patients (30.6%), with patients from Malay ethnicity making up half of the LTSC attendees (50.4%). Most of the patients' educational levels were either unavailable in the documentation or unknown (70.8%) during the audit period. Most patients were unemployed or were retired (77.9%) upon diagnosis, with the majority of the patients staying with family (94.7%). The demography of the audited patients is presented in Table 2.

The mean age of having a stroke among patients who attended LTSC was 57.68 years ± 12.04, with males having strokes earlier than female patients (56 years vs. 61 years). Most of the patients (72.6%) attended LTSC following their first stroke episode. Ischemic stroke was the most common subtype of stroke (67.3%). As for the risk factors, the majority of

**Table 2. Demographic characteristics of patients (n: 113).**

|  | Frequency (n) | Percentage (%) |
|---|---|---|
| **Age** | | |
| • Mean (SD) | 64.82 (9.77) | |
| **Gender** | | |
| • Male | 68 | 60.2 |
| • Female | 45 | 39.8 |
| **Ethnicity** | | |
| • Malay | 57 | 50.4 |
| • Chinese | 50 | 44.2 |
| • Indian | 4 | 3.5 |
| • Others | 2 | 1.8 |
| **Education level** | | |
| • No formal education | 1 | 0.9 |
| • Primary school | 6 | 5.3 |
| • Secondary school | 11 | 9.7 |
| • College/University | 15 | 13.3 |
| • NA | 80 | 70.8 |
| **Occupational status** | | |
| • Fixed salaried employee | 12 | 10.6 |
| • Self-employed | 13 | 11.5 |
| • Unemployed/Retiree | 88 | 77.9 |
| **Living arrangements** | | |
| • Alone | 3 | 2.7 |
| • Family | 107 | 94.7 |
| • Friends | 1 | 0.9 |
| • Nursing home | 2 | 1.8 |

our patients were non-smokers (92.9%) but had risk factors of diabetes mellitus (60.2%), hypertension (94.7%), or dyslipidaemia (92%). Following a stroke episode, most had been referred to physiotherapy (93%) and occupational therapy (65%) but not to a neurorehabilitation program (70.8%) or speech and language therapy (57%). The clinical profiles are presented in Table 3.

The audit generally showed that adherence to service parameters was met in nine out of eleven criteria assessed for long-term stroke care processes, as illustrated in Table 4. However, two specific criteria in the Post-stroke Checklist did not meet the standard performance. These criteria involved the assessment of the patient's fatigue and the inquiry about access to community resources. Table 5 demonstrates that while the achievement of the target for BP and HbA1c aligned with the expected performance for this audit, it fell short in the case of LDL-C.

Analyses of clinical parameters of the initial and latest measurements for each parameter found that patients exhibited a decrease in systolic blood pressure (SBP), diastolic blood pressure (DBP), low-density lipoprotein cholesterol (LDL-C), and glycosylated hemoglobin (HbA1c) levels. Additionally, a statistically significant reduction was observed in DBP (4.15 mm Hg, $p < 0.05$) and LDL-C (0.30 mmol/L, $p < 0.05$). These findings are presented in Table 6. Regarding the comparison of smoking status (Table 7), there is a reduction of post-stroke patients who smoke, from 15 patients (13.3%) to only eight patients (7.1%) who smoked during the most recent follow-up. Notably, nearly half of the smokers successfully quit smoking, resulting in a statistically significant decrease in patients who continued smoking ($p < 0.05$).

**Table 3. Clinical profiles of stroke patients attending LTSC.**

| CLINICAL PROFILES OF PATIENTS (N = 113) | | |
|---|---|---|
| | **Frequency (n)** | **Percentage (%)** |
| **Age of stroke** | | |
| • Mean (SD) | 57.68 (12.04) | |
| **Smoking status** | | |
| • Yes | 8 | 7.1 |
| • No | 105 | 92.9 |
| **Stroke episode** | | |
| • First | 82 | 72.6 |
| • Subsequent | 30 | 26.5 |
| • Unsure | 1 | 0.9 |
| **Stroke sub type** | | |
| • Haemorrhagic | 23 | 20.4 |
| • Ischemic | 76 | 67.3 |
| • Mixed | 3 | 2.7 |
| • Unspecified | 11 | 9.7 |
| **Diabetes mellitus** | | |
| • Yes | 68 | 60.2 |
| • No | 45 | 39.8 |
| **Hypertension** | | |
| • Yes | 107 | 94.7 |
| • No | 6 | 5.3 |
| **Dyslipidaemia** | | |
| • Yes | 104 | 92 |
| • No | 9 | 8 |
| **Neurorehabilitation program** | | |
| • Yes | 33 | 29.2 |
| • No | 80 | 70.8 |
| **Physiotherapy** | | |
| • Yes | 93 | 82.3 |
| • No | 20 | 17.7 |
| **Occupational therapy** | | |
| • Yes | 65 | 57.5 |
| • No | 48 | 42.5 |
| **Speech & language therapy** | | |
| • Yes | 56 | 49.6 |
| • No | 57 | 50.4 |

## Discussion

### Background of LTSC

This audit proved that post-stroke patients receiving specific services within the premise of a primary care setting benefited from clinicians' high adherence to clinical guidelines, observed from the improved clinical parameters. This adds to new findings in long-term stroke care evidence, as historically, most studies had been examining long-term stroke care from general practitioners' perspectives [10–12] or have centered specifically on addressing rehabilitation needs [13]. In contrast to conventional approaches, our service implemented a unique service

**Table 4. Adherence of services parameters on post-stroke checklist (n = 113).**

| Criteria | Adhered,n (%) | *Standard (%) | Achievement |
|---|---|---|---|
| 1. Secondary prevention<br>   Medical advice/medications | 111 (98.2) | 50 | Achieved |
| 2. Physical symptoms assessment<br>   ADL<br>   Mobility<br>   Spasticity/stiffness/contracture<br>   Pain<br>   Incontinence | 113 (100) | 50 | Achieved |
| 3. Speech<br>   Communication | 106 (93.8) | 50 | Achieved |
| 4. Cognition assessment<br>   Memory<br>   Emotional functioning | 108 (95.6) | 50 | Achieved |
| 5. Life after stroke<br>   Leisure activities<br>   Driving<br>   Back to work | 108 (95.6) | 50 | Achieved |
| 6. Relationship<br>   Personal/family | 105 (92.9) | 50 | Achieved |
| 7. Fatigue | 49 (43.4) | 50 | Not achieved |
| 8. Other challenges | 78 (69) | 50 | Achieved |
| 9. Nutritional | 99 (87.6) | 50 | Achieved |
| 10. Financial<br>   Self/social welfare | 93 (82.3) | 50 | Achieved |
| 11. Community resources | 30 (26.5) | 50 | Not achieved |

*Standard performance level was set at 50% for this first audit conducted on long-term stroke services after discussion with topic expert.

known as the LTSC, which integrates both comprehensive stroke care, including rehabilitation components with the existing family medicine values to facilitate a smooth transition of care for stroke patients from hospital to home, thus bridging the gap between hospital and home settings. Transitioning from acute hospital management to care at home could be a daunting experience, especially for stroke patients who are experiencing new clinical manifestations and disabilities related to stroke, all in need of adjustments and guidance. In their review of the effective transition of care from hospital to home for stroke patients, Mountain et al.

**Table 5. Attainment of the target during the latest follow-up at LTSC.**

| Clinical parameters target | Attained, n (%) | *Standard (%) | Achievement |
|---|---|---|---|
| BP < 140/90 mm Hg | 73 (64.6) | 70 | Achieved |
| HbA1c ≤ 7% | 68 (70.8) | 30 | Achieved |
| LDL-C < 1.8 mmol/L<br>(Non-diabetic) | 12 (27.9) | 45 | Not Achieved |
| LDL-C < 1.4 mmol/L<br>(Diabetic) | 2 (3.1) | 45 | Not Achieved |

*Individualized standard performance level was based on the Quality Assurance (QA) indicator within the Malaysian National Diabetes Registry Report(7) and Al-Salti et al.(9).

**Table 6. Comparison between initial and latest clinical parameters of patients attending LTSC (n = 113).**

| Item | Mean | | Paired differences | | | | | t | p |
|------|------|------|------|------|------|------|------|------|------|
| | | | Mean | SD | CI-95% | | | | |
| | Initial | Latest | | | | Lower | Upper | | |
| SBP | 130.3628 | 127.0973 | 3.26549 | 32.89947 | | -2.86670 | 9.39768 | 1.055 | 0.294 |
| DBP | 77.7788 | 73.6283 | 4.15044 | 19.09432 | | 0.59142 | 7.70947 | 2.311 | 0.023 |
| LDL-C | 2.5420 | 2.2444 | 0.29755 | 0.88059 | | 0.12796 | 0.46714 | 3.479 | 0.001 |
| HbA1C | 7.1843 | 6.7329 | 0.45135 | 5.69960 | | -0.70349 | 1.60620 | 0.776 | 0.440 |
| Weight | 67.0732 | 67.5280 | -0.45488 | 5.28060 | | -1.61515 | 070540 | -0.780 | 0.438 |

*Paired t-test was used to analyse the comparison data.

emphasized the provision of a seamless and effective healthcare continuation from in-hospital management to health provision in the community [14]. This concept is the basis for services provided in LTSC, which provides comprehensive clinical services for stroke patients in the community, encompassing aspects such as secondary prevention, managing comorbidities and complications, assessing mood and psychological adaptation, and working with a multi-disciplinary team approach namely rehabilitation, dietitian, neurology and others [15]. This approach is particularly valuable in areas where access to specialized stroke care and social services is limited as it provides a viable alternative, or in some cases, the only available option for stroke patients in the community. By providing services within the primary care clinic, it is able to provide familiarity, stability, and continuity of care, thus improving the outcomes and quality of life for stroke survivors within their home environments. To achieve this, LTSC relies on the Post-Stroke Checklist as the standardized management for the clinic, which was audited in this study.

## Clinical profiles and risk factors

The clinical profiles of patients who received care at the LTSC facility revealed that the average age of stroke occurrence was 57.68 years ± 12.04. Although this age was below the global average for stroke, which was reported as 64.4 years [16], it corresponded to the recent mean age for stroke prevalence in Malaysia (54.5–62.6 years old) [17]. This finding mirrored the current scenario in Malaysia, where stroke affects a much younger population compared to western

**Table 7. Comparison between initial and latest smoking status when attending LTSC (n = 113).**

| Smoking status | LATEST | | Total n (%) |
|------|------|------|------|
| | No | Yes | |
| INITIAL | | | |
| No | 98 | 0 | 98(86.7) |
| Yes | 7 | 8 | 15(13.3) |
| Total | 105(92.9) | 8(7.1) | 113(100) |

**p = 0.016.
*McNemar test was used to analyse the comparison data.
**Chi-Square test.

countries. This could be due to the increase in cardio-metabolic diseases in Malaysia, specifically hypertension, diabetes, and dyslipidemia, which are known risk factors for stroke, again reflected in our audit, showing high prevalence of hypertension (94.7%), dyslipidemia (92%), and diabetes (60.2%). These three modifiable metabolic risk factors are among the world's top leading stroke risk factors [18], indicating the importance of secondary prevention, which is one of the vital measures implemented in the LTSC and as part of the service parameters in the LTSC Post-stroke Checklist assessed during the audit.

On a global scale, approximately 62% of all incident strokes are classified as ischemic [18]. Our study reported a similar figure of 67.3%, indicating a similar occurrence of ischemic strokes. In terms of source of referral, most patients were referred to physiotherapy (82.3%) and occupational therapy (57.5%) following a stroke. The location of LTSC in the Klang Valley area could indicate that people in this area have convenient and ample access to allied health services. As observed in our study, a high proportion of patients (94.7%) stayed with their families, thus providing a reliable support system for stroke patients, especially in terms of monitoring and home management of long-term stroke complications. By involving family members or caregivers as integral components of comprehensive care, this arrangement can serve as leverage for improved post-stroke management, especially nursing care management such as bathing, self-care, and medication management. This is in line with the recommendations by Canadian Stroke Best Practice [14], in which persons with stroke, their families, and caregivers should be assessed and prepared for transitions of care from hospital to home. Families and caregivers should also be equipped with information and knowledge of stroke care, provision of education, skills training, psycho-social support, awareness of and assistance in accessing community services and resources.

## Clinical outcome parameters

The audit results indicate that the target BP and HbA1c levels aligned with the predetermined standard's performances. Moreover, the patients experienced significant improvements in their systolic blood pressure (SBP), diastolic blood pressure (DBP), low-density lipoprotein cholesterol (LDL-C), and HbA1c levels. Specifically, a statistically significant reduction was observed in DBP by 4.15 mm Hg (p<0.05) and LDL-C by 0.30 mmol/L (p<0.05). These reductions indicate a positive impact on cardiovascular health and lipid management as secondary prevention strategies in a stroke-specific clinic. Additionally, there was a notable decrease in the percentage of smokers, which was also statistically significant (p<0.05). The reduction in smoking rates suggests an improvement in overall lifestyle and a positive step towards reducing the risk of stroke and other related health complications. These findings aligned with previous local studies [19,20], which demonstrated similar improvements in blood pressure control and triglyceride (TG) levels following long-term stroke care services driven by primary care. The consistency of these results across studies reinforces the effectiveness of comprehensive, primary-care-driven approaches in achieving positive outcomes in blood pressure control and lipid management for long-term stroke patients. Overall, the audit results highlight the possible role of clinicians' high adherence to service parameters outlined in the LTSC Post-stroke Checklist in improving various clinical parameters and reducing risk factors. These findings support the value of the long-term stroke care services provided in the primary care setting and provide a foundation for further enhancements and future evaluations.

## Limitations and strengths

The audited medical records were limited to post-stroke patients in a specific urban area within Klang Valley, Malaysia. This restricts the generalisability of the findings to other areas

with limited availability of healthcare facilities; nonetheless, it highlights the feasibility of providing specific stroke services in a general primary practice service. The audit was only retrospectively audited at the year of service provision, which may have overlooked some of the findings determined in the standardized checklist. Despite the limitations, to our knowledge, this was the first and only local clinical audit that assessed the quality of longer-term stroke care offered as a specific service in primary care clinics in Malaysia. Data obtained for this audit were robust, as it was from an organized stroke registry of the clinic, thus providing good-quality clinical information. The findings also highlighted the importance of a post-stroke checklist as a management tool in longer-term stroke care.

## Conclusion

This study highlights the feasibility of conducting a specific stroke service within a primary care clinic that benefits post-stroke patients, as shown by the exceptional adherence and observed improvements in various clinical parameters. These improvements signify enhanced overall health and well-being among the post-stroke patient population, demonstrating the effectiveness of following established clinical guidelines in their management and recovery. Furthermore, these findings may catalyze clinicians to consider incorporating long-term stroke services as a specialized offering within primary care premises. The success achieved in this specific service underlines the potential benefits of on long-term stroke care within primary care settings. By establishing such specialized services, healthcare providers can ensure more comprehensive and tailored care for post-stroke patients, improving patient outcomes and enhancing quality of life. In future clinical audits, we propose including drug prescriptions as one of the parameters to be assessed in long-term stroke management. This enables us to assess the correlation between adherence and achieving optimal therapy status in stroke patients. To enhance service delivery, we recommend mandatory inclusion and emphasize the utilization of the LTSC Post-stroke Checklist as an integral consultation tool. Incorporating this checklist is practical and valuable for identifying the long-term care requirements of stroke patients within a clinical practice setting [21].

## Supporting information

**S1 Appendix. Check-list form used in the audit.**
(DOCX)

## Author Contributions

**Conceptualization:** Gadaffi Mostapha, Noor Azah Abd Aziz, Mohd Fairuz Ali.

**Data curation:** Gadaffi Mostapha, Noor Azah Abd Aziz.

**Formal analysis:** Gadaffi Mostapha, Noor Azah Abd Aziz, Mohd Fairuz Ali.

**Investigation:** Gadaffi Mostapha.

**Methodology:** Gadaffi Mostapha, Noor Azah Abd Aziz.

**Project administration:** Noor Azah Abd Aziz, Mohd Fairuz Ali.

**Resources:** Gadaffi Mostapha, Noor Azah Abd Aziz, Mohd Fairuz Ali.

**Software:** Gadaffi Mostapha.

**Supervision:** Noor Azah Abd Aziz, Mohd Fairuz Ali.

**Validation:** Gadaffi Mostapha, Noor Azah Abd Aziz.

**Visualization:** Gadaffi Mostapha.

**Writing – original draft:** Gadaffi Mostapha, Noor Azah Abd Aziz.

**Writing – review & editing:** Noor Azah Abd Aziz, Mohd Fairuz Ali.

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
