## [Decision Letter · Decision Letter 0]

9 Feb 2024

PGPH-D-23-02435

A Clinical Audit on Longer-Term Stroke Management as A Specific Service in A Primary Care Setting: Assessing Adherence of Service and Clinical Parameters

Dear Dr. Aziz,

Thank you for submitting your manuscript to PLOS Global Public Health. After careful consideration, we feel that it has merit but does not fully meet PLOS Global Public Health’s publication criteria as it currently stands. Therefore, we invite you to submit a revised version of the manuscript that addresses the points raised during the review process.

We look forward to receiving your revised manuscript.

Kind regards,

Hossam M Ashour, Ph.D.

Academic Editor

Journal Requirements:

1. Please provide additional details regarding participant consent. In the ethics statement in the Methods and online submission information, please ensure that you have specified (1) whether consent was informed and (2) what type you obtained (for instance, written or verbal, and if verbal, how it was documented and witnessed). If your study included minors, state whether you obtained consent from parents or guardians. If the need for consent was waived by the ethics committee, please include this information.

Additional Editor Comments (if provided):

Please make sure to address all concerns of reviewers.

Reviewers' comments:

Reviewer's Responses to Questions

**Comments to the Author**

1. Does this manuscript meet PLOS Global Public Health’s publication criteria? Is the manuscript technically sound, and do the data support the conclusions? The manuscript must describe methodologically and ethically rigorous research with conclusions that are appropriately drawn based on the data presented.

Reviewer #1: Yes

Reviewer #2: Partly

Reviewer #3: Yes

2. Has the statistical analysis been performed appropriately and rigorously?

Reviewer #1: Yes

Reviewer #2: Yes

Reviewer #3: N/A

3. Have the authors made all data underlying the findings in their manuscript fully available (please refer to the Data Availability Statement at the start of the manuscript PDF file)?

Reviewer #1: No

Reviewer #2: Yes

Reviewer #3: Yes

4. Is the manuscript presented in an intelligible fashion and written in standard English?

Reviewer #1: Yes

Reviewer #2: No

Reviewer #3: Yes

5. Review Comments to the Author

Reviewer #1: This is an important report on auditing registers of health care service provision of patients with stroke. This comes in the context of the increasing burden of NCD globally and particularly in the Global South. Clinical audit is an important tool for improving the provision of healthcare and assuring patients and the community. There is a need for more reports of this kind.

Few issues

1. In the background in the 3rd line: what part of the population the prevalence reported apply to?

2. Methods - Statistical Section: please specify what is being compared with the t-test and McNemar.

3. Results 3rd: We discovered that 18% (29/161) of patients were removed due to missing data. This must be described and explained in the methods. Moreover, you remained with 113 patients, so 161 - (113 + 29) = 19. What happened to these 19?

- What kind of missing affects the 29? I am very concerned that the remaining data has no missing data (except for education), making the whole situation really good. Missing/completeness of data should be a component of auditing as well therefore, unless clearly explained, those 29 should be included in the analysis.

4. Table 2:

- for age, indicate the unit of measurement, and report also the median and the IQR

- indicate that these characteristics were collected at entry, correct?

5. Table 3:

- for the age of stroke, indicate the unit (years?), and report also the median and the IQR

- Merge table 2 and table 3. There are so many tables here. I think PGPH allows a maximum of 5 tables.

6. Table 5:

- For blood pressure (BP < 140/90 mmHg), there is a 64.6% below the standard 70%. Why is this considered achieved? I am not disagreeing/agreeing I suggest clarifying the criteria for such a decision in the methods.

7. Table 6:

- this analysis requires a cohort followed. A hallmark of a cohort is follow-up time. The reported mean changes here suggest that the amount of follow-up time for every patient is similar. Is this a valid assumption? For example, one patient may take 6 months to reach the same change as another would require just 3. So, a change per month analysis would be better.

8. Table 7:

- Good that authors conducted a McNemmar test (the issue of follow-up time is OK)

- You could report an odds ratio. Add 0.5 in all cells to deal with the zero.

9. In the abstract, remove the p-values. Just put the confidence intervals.

10. In the discussion:

- remove the p-values. If you want to keep them, please report accurate p-values, not just p<0.05.

- The first and second paragraphs disregard an important bias when comparing cross-sectional. For your sample, you included only people who survived to be included in your sample. If those who have a stroke and die tend to be older than the ones in your sample, then you may have a biased mean age at stroke. Please reformulate your statement.

11. Have a list of abbreviations, please.

Reviewer #2: In the article, Mostapha et al. performed retrospective audit and published dataset to investigate the services for longer-term stroke care compared to the established standardized criteria. This article is not suitable for publication in its current form for the following reasons:

1. The objectives need more precise definition, and the paper's logical flow requires improvement. In the introduction, authors should clarify the existing gaps in the field, explain how their research addresses these deficiencies, and potentially fills the gap of unmet needs. Subsequently, each subsection within the results section should commence with a clear statement of purpose, ensuring a seamless transition from the preceding section. Consistent data interpretations and conclusions for each segment are essential within the results section.

2. The article contains a few grammatical errors and it would benefit from language editing.

3. The methods lack detailed descriptions. Figure legends are missing crucial information, such as the type of samples and experiments conducted.

4. The study involves the utilization of various databases and algorithms. It is essential for the authors to provide concise descriptions of each resource, outline the objectives of the analysis, and justify the reasons behind choosing these particular resources and methodologies.

Additional points:

1. The introduction should be served as a review of what is known prior to this current work. I suggest reviewing on current standardized tools, unmet needs of long-term stroke patient management and identifying risk factors of post-stroke complications. What are the current diagnosis limitations and challenges and how your work can help close the knowledge gap? I suggest listing some specific examples. Highlighting these topics will show the importance of your work.

2. The authors should seek for better representations for some of the figures, since some are not informative (see more comments below). The font sizes are too small in some figures.

3. For results, table 4 and 5, should be presented in better ways. For example, you can show adherence of services parameters on Post-stroke Checklist using bar plots and arrange by descending order.

1. For table 6, I suggest Please highlight the items that are significantly different when comparing initial and latest clinical parameter. Is P<0.05 the cut off for significance? I noticed weight is also one parameter, does weight loss indicate better post-stroke care or weight gain? Please specify this in the result sections.

5. I suggest include a flow chart for data selection.

6. In writing the result section, the authors should always provide some transitions from the previous section, and state their objectives first. For example, “having seen X, we next set out to understand the difference between initial and latest clinical parameters of patients attending LTSC. We perform Y, which showed Z.”

7. It would be helpful to include some future directions based on current study. For example, how would you expect to improve adherence to long-term stroke care. What are the current limitations for not adopting this care service vs. standardized?

Reviewer #3: The authors discuss the clinical audit on long term stroke management to assess adherence and clinical parameters

Please clarify the main goals of the study. Methods need more details. The manuscript needs some language editing to make it clearer. The authors need to discuss the results in light of the broader findings in the area.

6. PLOS authors have the option to publish the peer review history of their article (what does this mean?). If published, this will include your full peer review and any attached files.

**Do you want your identity to be public for this peer review?** For information about this choice, including consent withdrawal, please see our Privacy Policy.

Reviewer #1: **Yes: **Orvalho Augusto

Reviewer #2: No

Reviewer #3: No

---

## [Decision Letter · Decision Letter 1]

31 Oct 2024

A Clinical Audit on Longer-Term Stroke Management as A Specific Service in A Primary Care Setting: Assessing Adherence of Service and Clinical Parameters

PGPH-D-23-02435R1

Dear Prof Dr Aziz,

We are pleased to inform you that your manuscript 'A Clinical Audit on Longer-Term Stroke Management as A Specific Service in A Primary Care Setting: Assessing Adherence of Service and Clinical Parameters' has been provisionally accepted for publication in PLOS Global Public Health.

Best regards,

Hossam M Ashour, Ph.D.

Academic Editor

Reviewer Comments (if any, and for reference):

Reviewer's Responses to Questions

**Comments to the Author**

1. If the authors have adequately addressed your comments raised in a previous round of review and you feel that this manuscript is now acceptable for publication, you may indicate that here to bypass the “Comments to the Author” section, enter your conflict of interest statement in the “Confidential to Editor” section, and submit your "Accept" recommendation.

Reviewer #2: All comments have been addressed

Reviewer #3: All comments have been addressed

2. Does this manuscript meet PLOS Global Public Health’s publication criteria? Is the manuscript technically sound, and do the data support the conclusions? The manuscript must describe methodologically and ethically rigorous research with conclusions that are appropriately drawn based on the data presented.

Reviewer #2: Yes

Reviewer #3: Yes

3. Has the statistical analysis been performed appropriately and rigorously?

Reviewer #2: Yes

Reviewer #3: (No Response)

4. Have the authors made all data underlying the findings in their manuscript fully available (please refer to the Data Availability Statement at the start of the manuscript PDF file)?

Reviewer #2: Yes

Reviewer #3: (No Response)

5. Is the manuscript presented in an intelligible fashion and written in standard English?

Reviewer #2: Yes

Reviewer #3: Yes

6. Review Comments to the Author

Reviewer #2: The authors have addressed all the provided comments effectively. The objectives are now clearly defined in the introduction, with a logical flow that improves the coherence of the paper. The gaps in the field are articulated, and the authors explain how their research contributes to addressing unmet needs. Each results subsection now begins with a clear statement of purpose, providing seamless transitions and consistent data interpretation.

Grammatical errors have been corrected, and the language has been edited for clarity. Additionally, the figures have been updated for better representation, with improved font sizes. Tables 4, 5, and 6 have been revised for better presentation and clarity. Future directions based on the study have also been discussed, making the article more comprehensive. The manuscript is now suitable for publication.

Reviewer #3: (No Response)

7. PLOS authors have the option to publish the peer review history of their article (what does this mean?). If published, this will include your full peer review and any attached files.

**Do you want your identity to be public for this peer review?** For information about this choice, including consent withdrawal, please see our Privacy Policy.

Reviewer #2: No

Reviewer #3: No
